# Oxidative Stress and NRF2/KEAP1/ARE Pathway in Diabetic Kidney Disease (DKD): New Perspectives

**DOI:** 10.3390/biom12091227

**Published:** 2022-09-02

**Authors:** Daniela Maria Tanase, Evelina Maria Gosav, Madalina Ioana Anton, Mariana Floria, Petronela Nicoleta Seritean Isac, Loredana Liliana Hurjui, Claudia Cristina Tarniceriu, Claudia Florida Costea, Manuela Ciocoiu, Ciprian Rezus

**Affiliations:** 1Department of Internal Medicine, “Grigore T. Popa” University of Medicine and Pharmacy, 700115 Iasi, Romania; 2Internal Medicine Clinic, “Sf. Spiridon” County Clinical Emergency Hospital Iasi, 700111 Iasi, Romania; 3Department of Rheumatology and Physiotherapy, “Grigore T. Popa” University of Medicine and Pharmacy, 700115 Iasi, Romania; 4I Rheumatology Clinic, Clinical Rehabilitation Hospital, 700661 Iasi, Romania; 5Department of Morpho-Functional Sciences II, Physiology Discipline, “Grigore T. Popa” University of Medicine and Pharmacy, 700115 Iasi, Romania; 6Hematology Laboratory, “St. Spiridon” County Clinical Emergency Hospital, 700111 Iasi, Romania; 7Department of Morpho-Functional Sciences I, Discipline of Anatomy, “Grigore T. Popa” University of Medicine and Pharmacy, 700115 Iasi, Romania; 8Hematology Clinic, “Sf. Spiridon” County Clinical Emergency Hospital, 700111 Iasi, Romania; 9Department of Ophthalmology, Faculty of Medicine, “Grigore T. Popa” University of Medicine and Pharmacy, 700115 Iasi, Romania; 102nd Ophthalmology Clinic, “Prof. Dr. Nicolae Oblu” Emergency Clinical Hospital, 700309 Iași, Romania; 11Department of Pathophysiology, Faculty of Medicine, “Grigore T. Popa” University of Medicine and Pharmacy, 700115 Iasi, Romania

**Keywords:** oxidative stress, NRF2/KEAP1/ARE pathway, diabetic nephropathy, diabetes mellitus, antioxidant therapy, diabetic kidney disease

## Abstract

Diabetes mellitus (DM) is one of the most debilitating chronic diseases worldwide, with increased prevalence and incidence. In addition to its macrovascular damage, through its microvascular complications, such as Diabetic Kidney Disease (DKD), DM further compounds the quality of life of these patients. Considering DKD is the main cause of end-stage renal disease (ESRD) in developed countries, extensive research is currently investigating the matrix of DKD pathophysiology. Hyperglycemia, inflammation and oxidative stress (OS) are the main mechanisms behind this disease. By generating pro-inflammatory factors (e.g., IL-1,6,18, TNF-α, TGF-β, NF-κB, MCP-1, VCAM-1, ICAM-1) and the activation of diverse pathways (e.g., PKC, ROCK, AGE/RAGE, JAK-STAT), they promote a pro-oxidant state with impairment of the antioxidant system (NRF2/KEAP1/ARE pathway) and, finally, alterations in the renal filtration unit. Hitherto, a wide spectrum of pre-clinical and clinical studies shows the beneficial use of NRF2-inducing strategies, such as NRF2 activators (e.g., Bardoxolone methyl, Curcumin, Sulforaphane and their analogues), and other natural compounds with antioxidant properties in DKD treatment. However, limitations regarding the lack of larger clinical trials, solubility or delivery hamper their implementation for clinical use. Therefore, in this review, we will discuss DKD mechanisms, especially oxidative stress (OS) and NRF2/KEAP1/ARE involvement, while highlighting the potential of therapeutic approaches that target DKD via OS.

## 1. Introduction

Diabetes mellitus (DM) is a complex multisystemic metabolic disease that affects more than half a billion people worldwide, with the headcount projected to rise by almost 50% in 2045 [1,2].

With hyperglycemia as one of its hallmarks, it is known for leading to complications in the heart, kidneys, eyes and blood vessels [3]. These conditions generate functional impairment and early mortality; thus, DM remains a serious public-health concern [4,5]. Diabetic nephropathy (DN) is one of the main disorders that develops in both type 1 (T1DM) and type 2 (T2DM) diabetes mellitus [6]. The latter represents the most common etiology of end-stage renal disease (ESRD); meanwhile, it is reported that almost a third of diabetic patients will eventually be diagnosed with DN [7,8]. One issue that was raised during the pandemic that began in the spring of 2020 was the impact of COVID-19 infection on patients with DM and its complications. Studies have shown that COVID-19 associated with diabetes-related kidney disease increases both the hospitalization rate and the risk of premature death, therefore, adding a new layer to this complicated disorder [9,10].

Albumin loss in the urine, morphological changes/glomerular injury, and a reduction in the glomerular filtration rate (GFR) conventionally define this type of nephropathy [5,6]. However, over the last few years, a particular form of diabetes-related chronic kidney disease (CKD), with a decline in the GFR without micro-/macroalbuminuria, has been identified in some patients. Therefore, the term Diabetic Kidney Disease (DKD) encompasses both classical DN, with albumin loss in the urine and renal function impairment, and a particular form of nephropathy, distinguished by the reduction in GFR without albuminuria [11,12].

DKD has the most fulminant progression of all the conditions that cause CKD [13]. Despite its multifactorial pathogenesis, with several aspects playing a pivotal role, including older age, high blood glucose, high blood pressure, inflammation, metabolic syndrome and ischemic heart disease, the specific pathophysiological processes that occur are still unclear [4,14]. Oxidative stress (OS) is incriminated to have a crucial role in the development of DKD [15,16]. It is defined by an alteration in the oxidant and/or antioxidant state [17]. The heart and kidneys occupy the first two places in terms of number of mitochondria and oxygen uptake [18]. At the mitochondrial level, excessive reactive oxygen species (ROS) are generated because of the hyperglycemic environment; meanwhile, the same metabolic state can affect the main antioxidant-protective mechanisms [19]. Therefore, targeting both branches of the oxidant/antioxidant ratio through various pathways, such as the Nuclear factor erythroid 2-related factor 2/Kelch-like ECH-associated protein 1/Antioxidant responsive element (NRF2/KEAP1/ARE) pathway, can prove to be an efficient approach in the prophylaxis and management of DKD [15].

Taking into account the important role of OS in the development of DKD, with the NRF2/KEAP1/ARE pathway as a major modulator of the oxidative state of the cell, focusing on this pathophysiological branch can prove to be the key to a better management of this problematic disease.

## 2. Materials and Methods

We conducted broad research about the impact of OS in DKD using literature published over the last five years, up until August 2022. The electronic databases used as sources for suitable articles related to our topic were The US National Library of Medicine (PubMed), Scopus and Google Scholar. Studies published between the 1 January 2018 and the 1 August 2022 were selected to avoid any outdated data. We used various combinations of the following keywords: “oxidative stress”, “NRF2/KEAP1/ARE2 pathway”,”NRF2”, “diabetic nephropathy”, “nephropathy”, “diabetes mellitus”, “antioxidant therapy”, “diabetic kidney disease”. Diverse article types such as randomized controlled trials, clinical trials, multicenter studies, guidelines, reviews and meta-analyses were included. Only full-text English publications were selected. Furthermore, eligible and relevant papers were obtained from the reference lists of the previously selected articles. Two reviewers (E.M.G. and M.I.A.) primarily screened the articles by title and abstract. Thenceforth, under the supervision of a third reviewer (D.M.T.), they proceeded to full-text evaluation. All the relevant information extracted from the selected articles is summarized in text form. Firstly, we sum up the newly found connection between OS and DKD, the involvement of hyperglycemia and inflammation pathways and also highlight the NRF2/KEAP1/ARE pathway as a major modulator of intracellular OS. Lastly, we focused on potential therapeutic options in DKD via OS.

## 3. Mechanism and Pathophysiology of Diabetic Kidney Disease

Intricate pathophysiological changes precede the detection of DKD [20]. An extensive gene map realized by studying the genome of 157 patients of European descent who have CKD of different etiologies, including DKD, revealed that the key pathophysiology pathways are the ones that modulate both the metabolism and inflammation, with NRF2 being at the core of both [21,22]. Further, OS is considered to be a major feature of CKD and it is of significant importance as it has many cross-links with other pathophysiological elements [22]. Furthermore, the renin–angiotensin system, tubuloglomerular feedback, hyperglycemia, hypertension, autophagy and inherited predisposition have all been incriminated to play a role in diabetes-related kidney damage [13,23,24,25].

As we can see, the pathogenesis of DKD is extremely intricate and well connected, with multiple links between different pathways, where one element can generate or amplify another branch of this large network [26]. Therefore, it is essential to understand the hubs between key mechanisms, as they function more as a unit than individually. Thus, moving forward, we will focus on some of the main components of the pathogenesis of DKD and, particularly, on their connection to OS.

### 3.1. The Involvement of Hyperglycemia and Inflammation

Hyperglycemia and inflammation play a vital role in generating microvascular complications in DM, including DKD, through a gamut of interconnected mechanisms [24,27]. Currently, several tools are used to diagnose and monitor DKD, including micro-/macroalbuminuria, eGFR and kidney biopsy, each with its advantages and disadvantages [20]. However, new inflammatory and/or hyperglycemic biomarkers are currently being investigated [20,24].

As it has been shown that an improved blood glucose management in DKD can prevent the development of microalbuminuria and delay the progression of kidney injury, the best clinical available biomarker for assessing the risk of DM-derived complications remains glycated hemoglobin (HbA1c) [22,28,29]. An HbA1c level above 7% was correlated with the development of microalbuminuria in DKD, while every rise in HbA1c by approximately 1% increases the risk of microvascular complications in diabetic subjects by almost 40% [29].

Hyperglycemia can induce hemodynamic abnormalities, morphological changes and may trigger the activation of multiple pathways, including the protein kinase C (PKC) pathway, the generation of fructose via polyol activity, the overuse of the hexosamine pathway and the enhanced production of advanced glycation end products (AGEs) [25,30,31]. These various pathways have, as a downstream endpoint, the accumulation of ROS and the domino effect they generate [27]. The interaction between AGEs and the receptor (RAGE) represents one of the key elements in the development of microvascular complications [32]. Lower AGE levels can delay the progression of diabetes-related kidney injury, while an excess of AGEs triggers ROS accumulation and inflammation, via NF-kb activation [27]. In a hyperglycemic environment, glyoxalase 1 (Glo1) activity is reduced in the kidneys, which increases the level of AGE’s precursors, known as methylglyoxal (MGO). After that, the AGEs formed from MGO are free to activate RAGE and generate its subsequent effects [32].

The Rho-kinase (ROCK) pathway represents another significant mechanism activated by hyperglycemia [33]. Rho-kinase activation is involved in the regulation of complex cellular processes while its inhibition in DKD leads to a decrease in albuminuria [34]. Blocking Rho-kinase has also been associated with a decline in OS biomarkers, such as 8-hydroxydeoxyguanosine (8-OhdG), further proving its role in the pathophysiology of DKD [30].

The mitochondria are significantly influenced by the alteration in the redox state in hyperglycemic conditions [18]. The biomarker for mitochondrial impairment is the accumulation of ROS, beyond antioxidant resources, thus, placing mitochondria in a key role in the pathogenesis of DKD [31]. Further, excess ROS due to hyperglycemia triggers mitochondrial DNA mutations, which, in return, further exacerbate OS by altering oxidative phosphorylation [31,35]. Therefore, the characteristic mitochondrial dysfunction found in DKD leads to the activation of several intracellular-protective mechanisms, including mitophagy, which further lead to the progression of kidney injury [31]. Likewise, even though DKD is not standardly included in the group of inflammatory kidney diseases, many key elements related to inflammation have been linked to the onset and progression of this disorder [20,26,36]. The main components involved in the modulation of DKD inflammation are various cytokines (interleukin 1,6,18, tumor necrosis factor-alpha (TNF-α)), chemokines (e.g., monocyte chemotactic protein-1 (MCP-1)), adhesion molecules (vascular cell adhesion protein 1 (VCAM-1), Intercellular adhesion molecule 1 (ICAM-1)), cells (macrophage, mast cells, dendritic cells) and signaling pathways [19,37,38]. Further, persistent and pathogen-free inflammation is detected in these patients, confirmed by kidney histological changes, which further proves its connection to DKD pathogenesis [20].

Among the new biomarkers, MCP-1 is noted to be related to both inflammation and morphological changes, such as tubular atrophy and glomerular injury [29]. It has been observed that even in the early stages of DKD, an increased MCP-1 urinary excretion can be detected, thus, highlighting its relevance [29,34]. Interestingly, MCP-1 has also been found to be connected to other parameters, such as nephrin expression and kidney injury molecule-1 (KIM-1), which are biomarkers of incipient glomerular damage [28,29,38]. Assessing the cytokine cluster, interleukin 1,6 and TNF-α are distinguished as important markers of DKD progression, further proving the role of inflammation in DKD pathophysiology [22].

Although OS is presumed to be a precursor of inflammation at DKD onset, they both form an endless loop, where each will trigger or exacerbate the other, resulting in structural/kidney damage [39,40,41]. For example, different stimuli, including hyperglycemia and OS, can generate an increase in chemokines; meanwhile, TNF-α can stimulate, by itself, the formation of ROS, thus, representing a direct link to an imbalance in the oxidant/antioxidant state [37,38]. Furthermore, the macrophage, another important characteristic in the inflammatory process, can produce ROS, which may additionally aggravate the kidney morphological changes [34]. Last but not least, excess ROS, regardless of its origin, can activate transcription factors (e.g., nuclear factor-κB) and can trigger further pathways, such as Janus kinase/signal transducers and activators of transcription pathway (JAK-STAT), which intensify the inflammatory process, therefore, strengthening the inter-relationship between OS and inflammation in the pathophysiology of DKD [26,34].

### 3.2. The Key Role of Oxidative Stress in Diabetic Kidney Disease

OS represents a perturbance in the antioxidant/oxidant ratio, in which the denominator gains the upper hand [42]. Maintaining OS at high levels for a long period generates a chain of pathological changes that leads to apoptosis as the final outcome [43]. As a key pathophysiological element in CKD, it is essential to understand both the pro-oxidant and antioxidant components in OS to identify possible therapeutic targets.

The pro-oxidant branch of OS is divided into a group formed by ROS and another one formed by reactive nitrogen species (RNS), such as nitric oxide(^•^NO) and peroxynitrite (ONOO^−^) [12,44]. Taking into account that the main feature of OS in DKD is the accumulation of ROS, we will focus further on this category [7,12].

ROS are toxic molecules that originate from oxygen. They have low stability and can trigger oxidation reactions that can lead to structural and functional changes in biological compounds [44]. ROS can be split into two categories: a free radical group, which has in its structure an odd number of electrons, and a non-radical group, which is formed by two free radicals that join their electrons [45]. The primary exponents of the ROS group are superoxide (O_2_^•−^), hydrogen peroxide (H_2_O_2_) and hydroxyl radicals (^•^OH) [5].

ROS are formed in the kidney cells with the involvement of certain enzyme clusters, such as NADPH oxidases (NOX), and of some intracellular structures, such as the endoplasmic reticulum, peroxysomes and mitochondria [31]. The latter is the primary source of ROS, which is the result of redox reactions that take place in the electron transport chain [12].

ROS are known to play a key part in various cellular processes, such as cell growth, differentiation, injury, aging and death. They can also generate DNA damage, influence hormone action, regulate ion channel functioning and glucose metabolism [42,46,47]. It is now recognized that the damage caused by ROS occurs when there is a surplus, while a low value of ROS has important physiological roles, including in normal kidney function [48]. Increased levels of ROS are caused either by higher production, a failure of antioxidant mechanisms or a conjunction of both elements [12]. Therefore, ROS accumulation, beyond antioxidant resources, generates OS, which, in turn, reshapes normal cellular function, causes alterations in the protein and lipid metabolism, impairs mitochondrial activity and activates apoptosis [31,48]. As a result, at a larger scale, OS triggers inflammation, kidney scarring and other pathophysiological changes, leading to a further reduction in GFR and progression of DKD [48].

Antioxidant systems are essential in balancing redox homeostasis. A main part of the antioxidant defense is performed by enzymes, such as thioredoxin (Trx), catalase (CAT), cytochrome c oxidase, superoxide dismutase (SOD) and glutathione peroxidase (GPx) [7,44]. Other nonenzymatic structures, including glutathione (GSH), vitamin C, vitamin E and carotenoids, complete the antioxidant defense [45]. Various biochemical reactions characterize the mechanisms in which these components battle ROS. For example, the disintegration of hydrogen peroxide is realized by three elements that work in chronological order, SOD, GSH system and CAT, with oxygen and water as a result [44,45]. Therefore, as each antioxidant element has a defined function in a specific cellular compartment, they are all essential in maintaining redox balance [7,44,45].

## 4. NRF2/ARE/KEAP1 Pathway: Components and Functions

NRF2 has a main role in decreasing OS, by generating the transcription of various proteins [49,50]. It is a representative of the basic-region leucine zipper (bZIP) cluster, more precisely of the Cap-n-Collar subclass, which includes several transcription factors present in different species, including mammals [51]. NRF2 is made up of seven domains, called Neh, numbered from 1 to 7, each with a particular role [52]. Neh1 represents the domain that includes the bZIP motif, which modulates the interaction between NRF2 and DNA, via the antioxidant response element (ARE) [53,54]. This process is aided by the involvement of the Neh1 domain in the dimerization reaction, being facilitated by small musculoaponeurotic fibrosarcoma protein (sMAF) participation [53]. Further, this domain is implicated in controlling the stability of the whole protein and in the translocation of NRF2 from the cytoplasm to the nucleus [51]. The Neh2 domain has two motifs that interact with KEAP1, one that binds stronger (ETGE) to the protein, the other weaker (DLG). Moreover, this domain plays an important role in regulating NRF2 proteasomal degradation [55]. Neh3, Neh4 and Neh5 domains have a key function in triggering the gene transcription modulated by NRF2. Neh6 is involved in proteasomal degradation of the whole protein via two motifs (DSGIS, DSAPGS). This process does not include the participation of KEAP1 [53,55]. Neh7 has a role in suppressing NRF2 activity, as it binds with the retinoid X receptor-alpha [51].

KEAP1 is an intracellular protein, part of the BTB-Kelch family [53,56]. Its structure contains five domains, all of which have a key role in modulating the activity of NRF2 [51]. In basal conditions, NRF2 is both formed and destroyed. KEAP1 interacts with Cullin3 (Cul3) E3 ligase, forming a complex that accelerates both the ubiquitination and the proteasomal breakdown of NRF2, therefore, having a negative effect on its levels [56,57,58]. It is important to mention that, as a result of this interaction, the NRF2-KEAP1 complex remains in the cytoplasm [51,59]. When different stimuli, including ROS and electrophilic stresses, disrupt the previously steady environment, KEAP1 suffers an alteration in the reactive cysteine residues, generating a decrease in the enzymatic activity of the KEAP1-Cul3 structure [57,58]. After that, the new NRF2 that is produced after dissociation can enter the nucleus and can generate an NRF2-sMAF complex via a dimerization reaction that involves the Neh1 domain. This novel heterodimer complex binds to ARE, which increases the expression of several cytoprotective genes that are under NRF2 regulation [47,51,55]. As a result, this bond can generate a decrease in OS [7,17]. Li et al. [60] showed that in the NRF2 structure, a motif also exists that can detect, by itself, the alteration in cell reduction–oxidation homeostasis triggers, independent of the relationship with KEAP1; however, further studies are required to confirm this result.

When the NRF2/KEAP1/ARE pathway is triggered by increased ROS levels, multiple genes are activated and the production of several enzymes is enhanced, including CAT, NAD(P)H: quinone oxido-reductase1 (NQO1), SOD, heme oxygenase-1 (HO-1) and GPx. These enzymes eliminate, very effectively, intracellular ROS and corrupted cellular components, thus, decreasing inflammation and oxidative stress [5,7,31,61,62]. NRF2 also impacts the glutathione system and GSH synthesis by enhancing the production of glutamate-cysteine ligase (GCL), the rate-limiting enzyme in this biochemical process [5,7]. Furthermore, NRF2 acts not only through boosting the antioxidant defense mechanism, but also via decreasing the production of ROS via the advanced glycation end products (AGEs) [7]. Other than the effect on the redox state, NRF2 can decrease inflammation by interfering with the activity of nuclear factor-κB [22,51]. It is important to note that further research is needed to identify the link between NRF2 and cytokines, as so far, it has not been widely examined [63]. Last but not least, NRF2 also impacts the lipid and carbohydrate metabolism, interferes with mitochondrial functions and boosts insulin action [19,22].

As we can see, the NRF2/KEAP1/ARE pathway is a key and complex structure involved in DKD pathophysiology, with a major role in decreasing OS (Figure 1) and, therefore, it raises particular therapeutic opportunities in the management of DKD.

Arterial hypertension (HTA); renin-angiotensin-aldosterone system (RAAS); interleukin (IL); tumor necrosis factor-alpha (TNF-α); transforming growth factor β (TGF-β); monocyte chemotactic protein-1 (MCP-1); vascular cell adhesion protein 1 (VCAM-1); intercellular adhesion molecule 1 (ICAM-1); plasminogen activator inhibitor 1 (PAI-1); vascular endothelial growth factor (VEGF); Kelch-like-ECH-associated protein 1 (KEAP1); intervening region (IVR); endoplasmic reticulum (ER); reactive oxygen species (ROS); antioxidant response elements (ARE); superoxide dismutase (SOD); glutathione S-transferases (GSTs), catalase (CAT); heme oxygenase-1 (HO-1); glutamylcysteine synthetase (GCS); estimated glomerular filtration rate (eGFR); nuclear factor-κB (NF- κB); reactive nitrogen species (RNS); nuclear factor erythroid 2-related factor 2 (NRF2); endoplasmic reticulum (ER); oxidative stress (OS).

## 5. Therapeutic Approaches in Diabetic Kidney Disease via Oxidative Stress

Current treatment for DKD includes drugs targeting, on the one hand, the renin-angiotensin-aldosterone system and, on the other hand, the carbohydrate and lipid metabolism, with its share of side effects and without spectacular results, neither in improving renal function nor in stalling the natural evolution of the disease [8,39]. A novel therapeutic approach, including sodium glucose cotransporter 2 inhibitors (SGLT2) and glucagon-like peptide 1 receptor agonists (GLP1), can delay the progression to ESRD stage, independent of the hypoglycemic effect, but does not prevent its natural course [11,64]. For these reasons, there is a real need for drugs that target other pathophysiological mechanisms. Hence, this recent focus on the NRF2/KEAP1/ARE pathway may bring key revolutionary and novel therapeutic DKD approaches.

Many of the current therapies produced by the pharmaceutical industry and used in the medical field have been obtained by processing plants [65]. Therefore, the identification of new natural-derived molecules with therapeutical prospects represents the main interest of the research community. Moving on, we will concentrate on the main studied compounds that target DKD via the activation of the NRF2/KEAP1/ARE pathway, evaluating the pros and cons of each option.

### 5.1. The Main NRF2 Activators

#### 5.1.1. Bardoxolone Methyl

One of the most studied NRF2 activators is bardoxolone methyl (BM), considered the most important synthetic triterpenoid, being first used in oncology research [12]. It has been reported in the literature under several names, such as the methyl ester of 2-cyano-3,12-dioxoolean-1,9-dien-28-oic acid (CDDO-Me) or RTA-402 [66]. Triterpenoids, such as oleanolic acid, are phytochemicals known in Asian culture for their many biological benefits, including inflammation improvement/control and an antitumoral effect [67,68]. BM maintains the same advantages but it has different effects, depending on the amount administered. In a low dose, it can target the NRF2/KEAP1/ARE pathway, by triggering the unbinding of the NRF2-KEAP1 complex and its downstream effects, while in a higher concentration, it can impact tumoral cells [67,69].

Studies showed that BM has benefits in the management of several cancers, including pancreatic adenocarcinoma, mantle cell lymphoma and anaplastic thyroid carcinoma [68,70,71]. Additionally, a phase 1 study [71] emphasized an unexpected improvement in the estimated GFR (eGFR) in all cancer patients, paving the way for further studies targeting CKD. On this note, a small-sample phase 2 open-label study showed a significant increase in the eGFR after 56 days of treatment with BM, while highlighting the safety of the drug [72]. The BEAM (Bardoxolone Methyl Treatment: Renal Function in CKD/Type 2 Diabetes) (NCT00811889) study, a double-blind, placebo-controlled phase 2 trial, evaluated a total of 227 patients with CKD and T2DM. They were randomized to either placebo (57) or bardoxolone methyl 25 mg (57), 75 mg (57) or 150 mg (56). In all BM groups, a rise in the eGFR was noticed, which began 4 weeks after the first administration of the drug. From 12 weeks onwards, the greatest improvement in GFR was achieved and kept up to 52 weeks. The result was maintained for another 4 weeks after cessation of treatment, probably due to the reduction in OS. A paradox noticed in this study was a rise in albuminuria. Muscle spasm, especially in the calf region, was the most common adverse effect reported. Diarrhea, nausea, the temporary and minor rise of alanine transaminase, weight loss and hypomagnesemia were also observed, but the authors concluded that the adverse effects have a low impact and, therefore, the safety of the drug was sustained [69].

These promising findings prompted the phase 3 double-blind BEACON (Bardoxolone Methyl Evaluation in Patients with Chronic Kidney Disease and Type 2 Diabetes Mellitus: the Occurrence of Renal Events) (NCT01351675) trial, which included the randomized administration of either placebo or BM to 2185 patients from different centers with CKD stage 4 and T2DM. The quantity administered was 20 mg, with a different formulation than in the BEAM study, but comparable to the 75 mg dose from the latter. The main objectives of this trial were to determine whether treatment with BM can impact both the progression to ESRD and the mortality due to cardiovascular events. The study was terminated around 9 months after the first administered dose. This sudden shut-down happened because of serious events noticed in 33% of the patients from the bardoxolone methyl group, compared to 27% in the placebo group. The events reported in the BM group included a higher number of lung infections, heart failure and an increased hospitalization rate because of stroke and myocardial infarction. Further, it seems that mortality of patients treated with BM was higher, regardless of the etiology [73].

Thus, several issues have been raised regarding both BEAM and the BEACON trials, raising concerns and questions in the prospect of using BM in the future treatment of DKD. One of these issues is the alleged rise in the eGFR. Tayek and Kalantar-Zadeh [74] dispute that the rise in the eGFR in the BEAM trial could be due to the potential muscle loss that occurs in patients suffering significant weight loss, similar to members in the BM group. This fact can lead to lower serum creatinine and, thus, to a false increase in the eGFR because of a mathematical formula issue. They sustained this allegation with the fact that muscle spasms can be a symptom of muscle loss and, as seen, this adverse event was the most frequent in the bardoxolone methyl group [74]. Another possibility of eGFR improvement in the BEAM trial could be through a different mechanism. BM is likely to cause afferent arteriolar dilatation, because of a structural resemblance to cyclopentenone prostaglandins and, as a result, a rise in the intraglomerular pressure and, consequently, in the eGFR. The excessive loss of albumin through urine sustains this possibility [75,76]. In the long term, the initial increased GFR would follow a downward trend and, thus, the sustained intraglomerular pressure would harm the kidneys. All in all, the full spectrum of adverse effects reported in the BEAM study could have been the cause of the unpleasant development in the BEACON trial [74].

Therefore, the “Cinderella story” of BM that promised a breakthrough in the treatment of DKD was temporarily ended. However, not all was lost and there is some good to be taken out of the BEACON trial, even if it was suddenly shut down. Fewer patients from the bardoxolone methyl group than the placebo group progressed to ESRD and the improvement in the eGFR noticed in the BEAM trial was present also in the BEACON study, keeping hope alive in the treatment of DKD via targeting the NRF2/KEAP1 pathway and emphasizing the need for a different perspective [73]. Hence, a few years later, the TSUBAKI (The Phase 2 Study of Bardoxolone Methyl in Patients with Chronic Kidney Disease and Type 2 Diabetes) trial emerged. This double-blind, placebo-controlled Phase 2 study took place in 36 hospitals, all located in Japanese territory and enrolled 40 patients with T2DM and CKD stages 3 and 4. There were some very strict inclusion criteria, which were the result of the preceding trials [77]. Further analysis of the BEACON study data revealed that patients treated with BM who were not previously hospitalized for heart failure and with B-type natriuretic peptide (BNP) levels less than 200 pg/mL had a comparable risk of developing heart failure to patients in the placebo group. Thus, in the TSUBAKI study, the absence of these criteria was eliminatory in the selection of candidates. The patients enrolled were randomized to receive BM and given 5 mg per day first. Taking into account individual tolerance to the drug, the dose was increased to 10 mg after 4 weeks and 15 mg after 8 weeks. Following 16 weeks of monitoring, the TSUBAKI study showed that BM improves inulin clearance, eGFR and creatinine clearance, proving that the effect of the drug is, in fact, on the GFR and not dependent on the creatinine metabolism. No serious adverse events have been reported and there have been zero cases of heart failure or death. Likewise, it countered the hypothesis of increased intraglomerular pressure generated by bardoxolone methyl by showing that the drug’s beneficial effect on the eGFR was preserved for 1 year at a minimum and no downward trend was noticed in these patients [77].

These results laid the ground for another randomized, double-blind, placebo-controlled Phase 3 study known as AYAME (A Phase 3 Study of Bardoxolone Methyl in Patients with Diabetic Kidney Disease) (NCT03550443), which enrolled more than 1000 subjects with DKD. The main objectives are to prove the safety and efficacity of bardoxolone methyl administered at a dose of 5, 10 or 15 mg. The trial is currently in progress and is predicted to be completed in the spring of 2024 [78].

Other recent trials that study BM efficacity and safety are the EAGLE trial [79], CARDINAL trial (targeting patients with Alport syndrome) [80,81] and FALCON trial (targeting patients with autosomal-dominant polycystic kidney disease (ADPKD)) [82]. Therefore, BM still represents a viable option in the future treatment of CKD, but it is yet unknown which CKD etiology will benefit from its effects.

#### 5.1.2. Curcumin and Analogues

Curcumin is one of the most important NRF2 activators [7,83]. Curcuma Longa Linn, of the Zingiberaceae family, contains a structure called rhizome, which is converted into a powder, known as turmeric, from which a natural component named curcumin is derived. Curcumin is part of the polyphenol group, known primarily for its use as a culinary spice [84,85,86]. It has many known biological roles that may influence the treatment of several oncological, neurological, cardiological and nephrological conditions [87]. Curcumin can modulate inflammation, the immune system, lipid and carbohydrate metabolism and prevent the development of DM-derived complications, including DKD [84,88]. It also plays a role in decreasing OS [7]. This effect, demonstrated in vivo [89,90] and in vitro [91,92], is obtained by breaking the bond between NRF2 and KEAP1, which triggers the activation of the NRF2/KEAP1/ARE pathway, resulting in the increased production of antioxidant enzymes [7,85,93]. Therefore, considering that OS is a major pathophysiological component in DKD, curcumin represents a great prospect in the treatment of this condition.

Curcumin has some disadvantages regarding its pharmacokinetics, which restrict its huge potential. Its chemical molecular structure is unstable, which decreases its bioavailability [94]. Further, curcumin has to be administered in high doses to exert its therapeutical roles. In the in vivo studies conducted on DKD animal models [95,96,97], the research team used curcumin 50–200 mg/kg per day, while in the clinical trials, reviewed by Jie et al. [84], the patients who received unaltered curcumin were given a dose between 320 and 1670 mg per day. Last but not least, its absorption is limited, while its degradation and elimination are fast, posing more challenges in the medical use of curcumin [98].

A recent systematic review and meta-analysis of a previous human, randomized, double-blind, placebo-controlled clinical trial [87] showed that curcumin improves serum creatinine levels, fasting blood glucose, systolic blood pressure and total cholesterol in DKD patients. The same authors also pointed out a drawback of this analysis; curcumin did not exert a significant impact on proteinuria, blood urea nitrogen and other parameters, supporting the need for further improved studies, with more patients [87].

This is why the focus is shifting to other strategies in curcumin research. One of them is the production of adequate derivatives that maintain bioactivity but have improved pharmacokinetics [94]. An example of an analogue obtained from curcumin is the molecule known as C66-curcumin((2E,6E)-2,6-bis [2-(trifluoromethyl)benzylidene]cyclohexanone) [99]. Wu et al. [100] used C66-curcumin 5 mg/kg once every two days in vitro. The drug administration represents an improvement in the original curcumin scheme. The authors showed that the C66-curcumin effect on DKD is mediated by the NRF2/KEAP1/ARE pathway, but also highlighted an NRF2-alternate mechanism of action. This accessory function is a C66-curcumin particular feature that other NRF2 activators, such as sulforaphane (SFN) and cinnamic aldehyde (CA), do not possess [100]. Other analogues, such as J17 ((2E,5E)-2-(3-Hydroxy-4-methoxybenzylidene)-5-(2-nitrobenzylidene)cyclopentanone) and B6 ((E, E)-1, 5-bis(2-bromophenyl)- 1,4-pentadiene-3-one), might have positive effects in preventing and treating DKD, but this remains to be established by further studies [101,102].

#### 5.1.3. Isothiocyanates and the Main Representatives: Sulforaphane, Moringa Isothiocyanate

Isothiocyanates (ITCs) are natural components in cruciferous vegetables, such as broccoli, cabbage, cauliflower and Brussels sprouts. They are rich in glucosinolates (GSL), including glucoraphanin, and become bioactive after the hydrolysis of GSL mediated by the myrosinase enzyme [103,104]. This reaction occurs after damaging the plant by mechanical processes, such as cutting and chewing, and is inhibited by the exposure to the high temperatures used during cooking [105]. Even though myrosinase is absent in humans, a myrosinase-like effect can be traced in the intestinal microflora [106,107].

SFN or 1-Isothiocyanato-4-methyl sulfinyl butane is the most studied representative of the ITC family [108,109]. When administered in an average dose, SFN has a good safety profile and, despite having one of the best bioavailabilities in the phytochemicals group, it is also an unstable molecule, having some pharmacological drawbacks, such as large liver first-pass effect and decreased water solubility [65,105,106,108]. Several analogues have been evaluated but they have proved to be inferior to the original molecule [103,110].

The highest level of glucoraphanin is found in broccoli seeds and, thus, most of the studies conducted that analyzed SFN used broccoli-derived compounds [65,111]. The effects of SFN have been evaluated extensively in many disorders, including several cancers [112,113,114], liver diseases [115,116], neurological [110,117], ophthalmological [118] and cardiological [119,120] conditions. In DKD, SFN triggers the NRF2/KEAP1/ARE pathway and its role is entirely dependent on NRF2 activation [121]. It can also modulate inflammation, by interacting with NF-kb [106]. Many studies conducted in vivo, on different animal models [121,122], concluded that SFN can prevent or delay diabetes-related kidney damage, but the lack of clinical trials prevents any conclusions from being drawn [106,123]. Although some clinical studies of SFN in other conditions have been completed [124,125,126], a trial, including subjects with DKD, is needed to certify the safety and effect of SFN in this particular disease [108].

Another representative of the ITCs is Moringa isothiocyanate (MIC-1), extracted from Moringa oleifera [127]. MIC-1 maintains some of the beneficial effects of SFN, as it activates NRF2 and decreases OS, as was shown in a recent in vitro study [62]. However, it is yet unknown if MIC-1 could be a better alternative to SFN and further studies are required to evaluate this particular molecule.

#### 5.1.4. Cinnamic Aldehyde

CA, known also as cinnamaldehyde, is a compound derived from cinnamon, with many known beneficial effects [128]. The formulation of cinnamon oil, obtained from cinnamon bark, contains almost exclusively CA [129]. The main actions of this particular molecule are aimed at dealing with tumors, infections, inflammation and OS. CA can also impact metabolic diseases, such as DM, by improving glycemic control and its symptoms, including polyuria and polydipsia [130,131].

CA, as a chemical compound, raises some questions regarding its bioavailability, as it can be rapidly transformed into cinnamic acid [132]. Further, some side effects are possible, including skin allergic reactions and liver toxicity [132,133].

Most studies conducted in vitro [131,133] or in vivo [130,131,134,135,136] evaluate its impact on DM and some of its complications, which emphasize the beneficial role of CA in lowering OS via the activation of the NRF2/KEAP1/ARE pathway [136]. Unfortunately, there is a paucity of research conducted on DKD, which prevents the assessment of its possible beneficial effect on this particular disease [131,133,136].

#### 5.1.5. Resveratrol

Resveratrol (RES, trans-3,5,40-trihydroxystilbene) is part of the polyphenol group, found in different natural structures, such as peanuts, vegetables, fruits, cereals and wine [137,138]. RES’s popularity comes from its role as an effective antioxidant, anti-inflammatory, anti-aging and anti-diabetic agent. It possesses some of the pharmacological characteristics present in other natural compounds, such as decreased bioavailability, poor water solubility and fast metabolism, which need further improvement [139].

The spotlight nowadays is on its influence on DKD, which is mediated through several pathways, including the NRF2/KEAP1/ARE pathway. Thus, the main advantage of this molecule is highlighted, as it can target, at once, several pathophysiological mechanisms involved in the development and progression of DKD [137,139]. The impact on this condition is proven in vitro [140], in vivo [140,141,142,143] and also in small-sample clinical trials [144,145]. A recent systematic review and meta-analysis of animal studies [143] evaluating the effect of RES on DKD showed that several parameters are influenced by this molecule, including blood glucose levels and serum creatinine. An interesting finding was that no side effects were reported, but this outcome should be assessed with caution, as several factors could have had an impact, including dosages and a brief administration time of the drug [143]. Li et al. [139] pointed out that some of the data in the literature on RES are not as favorable, as some conflicting results appear, reinforcing the need for larger-sample studies and better assessment of the ideal dose and period of administration (Table 1).

### 5.2. Other Compounds That Modulate the NRF2/KEAP1/ARE Pathway

#### 5.2.1. Allicin

Allicin (thio-2-propene-1-sulfinic acid S-allyl ester) is a molecule that can increase the expression of NRF2, thus, improving intracellular redox balance. When garlic (Allium sativum L.) goes through a series of mechanical reactions, allicin formation is launched from a precursor named alliin [146]. Its antioxidant effect on DKD was proven by Arellano-Buendía et al. [4] in a study on diabetic rats, while other possible roles, such as wound repair [147] and anti-inflammatory [148] activity, have been recently noted in the literature.

#### 5.2.2. Grape Seed Proanthocyanidins Extract and Eucommia Ulmoides

Grape Seed Proanthocyanidins Extract (GSPE) is another representative of the polyphenol family [35]. GSPE is an oligomer naturally found in grape (Vitis vinifera) seeds that has shown many biological activities, including an antioxidant role [149]. GSPE can decrease OS in a greater capacity than other natural antioxidant compounds, such as vitamin E and C [150]. Its many actions were studied in several diseases and some promising results were obtained in DM and diabetic-related complications, such as diabetic bladder dysfunction [151], non-proliferative diabetic retinopathy [149] and diabetic peripheral neuropathy [152]. The studies conducted on DKD showed that GSPE can improve this disorder by targeting many pathological mechanisms, such as AGE/RAGE and nephrin expression [153,154]. The question was raised whether GSPE can also target the NRF2/KEAP1/ARE pathway. In vitro studies conducted on other compounds extracted from grapes showed that they can influence NRF2 activity, but GSPE’s effect remained unknown [155,156]. Ding et al. [157] showed that GSPE displayed a protective action on a rat model in DKD by triggering the activation of the NRF2/KEAP1/ARE pathway. This finding needs to be further investigated in other studies that can also evaluate other possible effects of GSPE in DKD.

Another compound that can activate NRF2 and target the AGE/RAGE pathway is Eucommia ulmoides, a natural plant frequently used therapeutically in Asian culture, but research on this molecule is currently limited [158].

#### 5.2.3. Other Polyphenols

Polyphenols represent a chemical category abundant in plants, characterized by the presence of at least one hydroxyl substituent and two phenyl rings. They can be divided into flavonoids and non-flavonoids [159]. Some of the representatives of this family, such as curcumin, RES and GSPE, have already been discussed in this article. Other compounds from this group have shown a reno-protective effect in DKD by targeting the NRF2/KEAP1/ARE pathway. One of them is Rutin, a flavonoid that displayed a beneficial role in diabetic-related kidney injury through several mechanisms, including as an NRF2 activator [160,161,162,163]. Further, Lespedeza bicolor and the roots of Cudrania tricuspidata Bureau (CTRE) are compounds rich in flavonoids and other phenol compounds that possess the ability to modulate inflammation and trigger the NRF2/KEAP1/ARE pathway, decreasing, as a result, intracellular OS [32,164]. Another studied flavonoid is Icariin, found in Epimedium plant species. It possesses many biological roles, while in DKD, it shows an anti-fibrotic and antioxidant effect, the last one in an NRF2 partially dependent pathway [165,166]. Other representatives that can trigger the NRF2/KEAP1/ARE pathway are Diphlorethohydroxycarmalol, derived from Ishige okamurae, a species of algae [167], Genistein [6], Hesperetin [168], Myricetin [169] and Digitoflavone (3,0,4,5,7-tetrahydroxyflavone) [170,171]. Further, some potential NRF2 activators that are part of the same family have been studied in DKD, with promising results, such as Auricularia cornea [172].

#### 5.2.4. Antioxidant Compounds Targeting Mitophagy

Coenzyme Q10 (CoQ10), a compound similar to vitamins, plays a major role in respiratory chain reactions [7]. In DM, CoQ10 was shown to have an antioxidant effect as it can increase NRF2 expression [173]. Sun et al. [18] further evaluated this role in DKD and showed that CoQ10 can activate the NRF2/KEAP1/ARE pathway, which, as a result, improves mitophagy, alleviating diabetic kidney injury. The same activity is present also in other compounds, such as MitoQ [174], which targets mitochondria, and Astragaloside II (AS II) [175], obtained from Astragalus membranes. As a result, all of these molecules can ameliorate normal cell homeostasis by improving the antioxidant/oxidant balance [7,174,175]

#### 5.2.5. Astaxanthin (AST)

AST is a xanthophyll of maritime origin, part of the carotenoid group [176]. The bio properties of AST are diverse, including modulating inflammation, cell death and diabetes, while an impressive safety profile completes its characteristics [177]. Its OS-lowering effect is 800-fold higher than the previously described CoQ10 [178]. AST’s action on redox balance in DKD is NRF2 dependent, as was shown by Chen et al. [179] and Zhu et al. [178] in their research. These aspects promote AST as an attractive antioxidant agent, with great potential.

#### 5.2.6. Drugs with NRF2 Activity

Some drugs that are currently in use in the medical field for other conditions have shown the ability to target the NRF2/KEAP1/ARE pathway. One of them is Fenofibrate (FF), which is known for its effect on lipid metabolism. FF displayed the ability to activate NRF2, with the assistance of fibroblast growth factor 21 (FGF21), thus, preventing further diabetic-related kidney injury. This finding is essential as FF is a well-known drug, already administered to patients worldwide, with a good general safety profile [15,180]. Further, Arellano-Buendía et al. [61] showed that mycophenolate mofetil (MMF), an immunosuppressant drug, can activate the NRF2/KEAP1/ARE pathway by lowering KEAP1 levels [61,181]. Another molecule that could have a reno-protective effect in DKD is minocycline, an antibiotic part of the tetracycline subclass. It has a particular mechanism of action, as it can impact redox homeostasis by reducing NRF2 ubiquitination [182]. All of these therapies are already reviewed in the literature; therefore, this novel effect they possess, if proven by further studies, could change their primary use.

#### 5.2.7. Less-Known NRF2 Activators

Other NRF2 activators are being studied but they are currently in an early stage of research. Thus, they require further evaluation to be considered as potential future therapies for DKD treatment. Sodium butyrate (NaB) and AB38b are two synthetic compounds evaluated in diabetic mouse models that could be effective agents in improving DKD via the NRF2/KEAP1/ARE pathway [183,184]. Another molecule that can activate NRF2 and decrease OS in DKD is Tetrandrine, an alkaloid derived from Stephania tetrandra [40]. Further, in a recent study conducted by Zhang et al. [185], a compound found in Alpinia officinarum, known as (R)-5-hydroxy-1,7-diphenyl-3-heptanone (DPHC), demonstrated an antioxidant effect in vivo and in vitro by triggering the NRF2/KEAP1/ARE pathway. Last but not least, many other NRF2 activators, such as Berberine [186] and Broussonetia kazinoki Siebold fruit [187], are documented in the literature because they displayed reno-protective activity in DKD.

As observed, there are completed and currently ongoing trials that investigate novel molecules and compounds, which exert antioxidant effects via regulation of the NRF2/KEAP1/ARE pathway and, hence, hold potential as new therapeutic targets in DKD management (Table 2).

Although, the eGFR, urea, creatinine, ACR (the urinary albumin-to-creatinine ratio), 24 h urine parameters, glycemic profile and other renal function parameters are standardly detected in easily obtained blood and urine samples in both human and experimental studies, when it comes to the quantification of OS as a target, we are restrained by a specific method of detection. In the majority of human clinical studies, the assessment of the therapeutic-OS response is not clearly defined, while in pre-clinical studies, these changes were seen in samples of kidney tissue (histology examination) and cell cultures, using different staining methods and assays (e.g., Western blotting, real-time polymerase chain reaction). This emphasizes the need to develop a specific assay for OS-therapy detection and monitoring that can be used in clinical practice along with a more complex and complete panel of OS biomarkers.

Overall, NRF2 activators and natural compounds that target OS show, so far, beneficial results in both preclinical and clinical studies. However, there are limitations to their use that need to be discussed, which explains why they are not currently implemented in clinical practice. For instance, clinical trials involving BM, one of the most promising targets, raised the need for newer longer-term data that may provide the evidence required to determine its risks and benefits. Although there are many other compounds, such as curcumin, sulforaphane, resveratrol and other natural molecules in the form of low-priced dietary supplements, due to a lack of high-quality research and complete larger human trials, the Food and Drug Administration (FDA) and experts do not recommend their use for disease prevention. Additionally, other potent molecules that target OS, because of their reduced bioavailability, inferior water solubility and delivery issues, have limited pharmaceutical potential for further application. Despite, their more expensive profile NRF2 activators exert DKD-promising results, which raises the need for further larger clinical trials and clearer results, which can promote their potential use for wide clinical application.

After meticulous research of the most relevant reviews conducted in this field, we have to mention some of the differences regarding our paper. For instance, in their work, Stenvinkel et al. [22] discussed the mechanisms of chronic inflammation in CKD, with a focus on NRF2, without any mention og treatment options. Behl at al. [19] focused on mechanisms and involvement of NRF2 but strictly in DM, while therapy is briefly mentioned. Ito et al., in their review, pointed out the role of NRf2 in CKD [50]. Guerrero-Hue and colleagues [55] also focused on NRF2 as a therapeutic option and discussed the known clinical trials of NRF2 inducers in CKD of different etiologies (e.g., CKD, ADPKD and Alport, CKD with T1DM/T2DM/IgA nephropathy). Adelusi et al. [188] only covered the implications and the therapeutic potential of KEAP1/NRF2/ARE in DKD, while Sakashita [12] described briefly, in addition to the mechanism, OS as a therapeutic target, with a focus only on Bardoxolone Methyl. This recent review [189] includes data about diabetic neuropathy and their section about therapy is shorter and does not include other OS targets. Dragoș et al. [39] and Tang et al. [190], in their papers, highlighted the mechanisms of herbal nephroprotection in diabetes mellitus and diabetic nephropathy, respectively.

As observed, we believe that our up-to-date manuscript incorporates scientific data not only about the pathophysiology behind DKD, involving the intricate relationship between hyperglycemia, inflammation and OS and the role of NRF2 in these processes, but also covers a larger spectrum of information on antioxidant therapy. Additionally, our review highlights both preclinical and clinical data about the main NRF2 activators and integrative research about other antioxidant compounds and less-known NRF2 activators. The purpose of this extent is to highlight how broad this field is, with still unknown pathways and molecules with biomarker potential and not-so-discussed potential therapeutic targets, with the hope that it can help future researchers.

## 6. Conclusions

DKD is a big challenge for the research field, as there is currently no optimal treatment available for this complicated disease. Drugs targeting OS via the NRF2/KEAP1/ARE pathway can offer a solution to this problem. As we can see, there is a plethora of options to choose from when looking for possible future and revolutionary DKD therapy. Such a large number of alternatives can be overwhelming, as it can lead to a dispersed allocation of financial and human resources that hinders a rapid outcome from being achieved. Therefore, the focus of the research field in DKD should be on conducting large clinical trials that evaluate the action of the most promising molecules that target the NRF2/KEAP1/ARE pathway, paving the way for an important breakthrough that can alleviate this heavy burden.

## Figures and Tables

**Figure 1 biomolecules-12-01227-f001:**
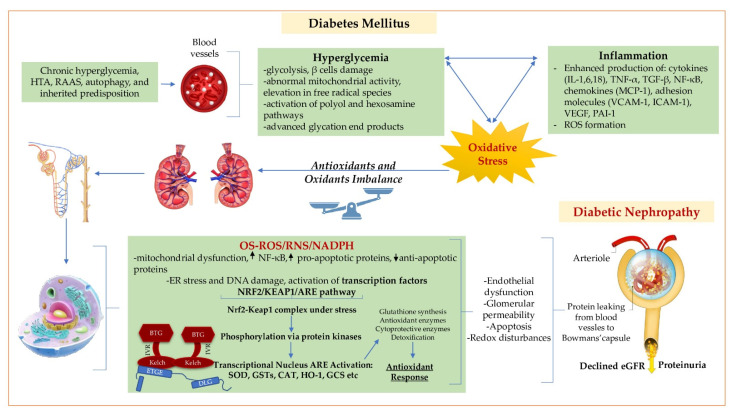
Role of oxidative stress and NRF2/KEAP1/ARE pathway in the multiplex diabetic nephropathy pathophysiology. DM associated with multifarious risk factors through persistent hyperglycemia, inflammation and OS, can instigate renal integrative glomerular barrier complex dysfunction (IGBC). IGBC is mainly formed by endothelial cells, the glomerular basement membrane (GBM), podocytes, tubular epithelial and mesangial cells. Each of these cells under influence of the disruptive mechanism mentioned above, produce factors and activate intricate pathways that have individual or entangled repercussion, which finally leads to impairment of the renal filtration unit. The major targets of OS are the fenestrated glomerular endothelial cells and the podocytes cells. Here, ER and mitochondrial stress enhances transcriptional activation of the NRF2/KEAP1/ARE pathway which can lead to a beneficial antioxidant response.

**Table 1 biomolecules-12-01227-t001:** Key points of the most relevant NRF2 activators and future view of their use in CKD.

NRF2 Activators and Ref.	Strengths	Weaknesses	Future Perspectives
Bardoxolone methyl [69,73,77,78,80,82]	GFR ↑	Albuminuria ↑	Its strengths could be present without its weaknesses in subjects who: - have not previously been hospitalized for heart failure and with BNP levels less than 200 pg/mL -other etiologies of CKD (ADPKD, Alport syndrome)
	Delayed progression to ESRD	Morality↑ and serious side effects in some groups	
Curcumin [84,94,98]	Serum creatinine levels ↓	No significant impact on proteinuria, blood urea nitrogen	Curcumin derivatives could be a better option from a pharmacokinetic point of view while maintaining the beneficial effects of the original compound
	Fasting blood glucose ↓	Unstable molecular structure	
	Systolic blood pressure ↓	High dosages to obtain the known effects	
	Total cholesterol ↓	Absorption ↓	
		Degradation and elimination ↑	
Sulforaphane [65,105,106,108]	Good bioavailability	Unstable molecular structure	A clinical trial could bring a better understanding of its future potential
		Liver first-pass effect↑	
		Water solubility ↓	
Cinnamic aldehyde [130,131,132,133]	Improves DM symptoms	Bioavailability?	There is less data available on its effect on DKD, which highlights the direction future studies need to take.
		Possible side effects	
Resveratrol [137,139]	It can target at once several pathophysiological mechanisms	Bioavailability ↓	There are some conflicting results, which need to be evaluated in further studies
	Possible no significant side effects?	Metabolism ↑	
		water solubility ↓	

raised ↑; lower ↓; B-type natriuretic peptide (BNP); autosomal dominant polycystic kidney disease (ADPKD); Diabetes mellitus (DM); Glomerular filtration rate (GFR); End-stage renal disease (ESRD); Chronic kidney disease (CKD); Diabetic kidney disease (DKD).

**Table 2 biomolecules-12-01227-t002:** Novel compounds with therapeutic potential in DKD via modulation of the NRF2/KEAP1/ARE pathway.

Compound	Animal/Cells	General Effects	Doses and Time of Administration	Detection Site and Notable Findings	Ref.
Allicin	Male Wistar rats	antihypertensive, antidiabetic, antioxidant, antifibrotic effects	16 mg/kg day/p.o. for 30 days	- kidney tissue: increased Nrf2 expression and decreased SBP, Keap1, HIF-1α, and VEGF expression. - kidney tissue: increased nephrin, KIM-1, the mesangial matrix, fibrosis index, and the fibrotic proteins, hyperglycemia. - skeletal muscle: improved insulin levels, and prevented changes in (GLUT4) and IRSs.	[4]
GSPE	STZ-induced diabetic rats	antioxidant effects, can decrease insulin resistance, delay DKD progression and improve DKD	I group (treated with 125-mg/kg/day GSPE for 8 weeks), and II group (treated with 250 mg/kg/day GSPE for 8 weeks)	- kidney tissues: significantly increased the levels of total antioxidative capability, and glutathione (*p* < 0.05), protein levels of Nrf2, HO-1, GST, and NAD (P)H quinone oxidoreductase 1 (*p* < 0.05)	[157]
EU	STZ-induced diabetic mice	antioxidant, anti-hypertensive, and anti-hyperglycemic effect	(200 mg/kg) orally for 6 weeks	- kidney tissue and plasma: significantly upregulated Nrf2 expression but downregulated that of receptor for AGE (RAGE). - plasma: ameliorated the renal damage by reducing OS, via the Glo1 and Nrf2 pathways.	[158]
Rutin	HRGECs	direct antioxidant effect on human renal glomerular endothelial cells	12.5, 25, or 50 µM rutin and/or HG for 24 h	- renal endothelial cells: significantly prevented hyperglycemia-disrupted renal endothelial barrier function by inhibiting the RhoA/ROCK signaling pathway via lowering ROS, that was mediated by the activation of Nrf2.	[160]
CTRE	HK-2 cells	antioxidant, antiinflammatory effects	(5–40 μg/mL), ranging between 3 and 24 h	- renal proximal tubular cell: alleviated the methylglyoxal (MGO)-induced decrease in nuclear factor (erythroid-derived 2)-like 2 (Nrf2), inhibited induction of NADPH oxidase 4 (NOX4)	[164]
Icariin	HMC and/or STZ-induced diabetic rats	antioxidant effect, can prevent the development of DKD, improve DKD-induced kidney injury	(20, 40, 80 mg/kg, i.g.) group for 9 weeks	- kidney tissues: decreased the levels of intracellular superoxide anion, increased dissociation of Nrf2/Keap1 complexes, Nrf2 translocation to nuclear, Nrf2/ARE DNA binding activity, and ARE luciferase reporter gene activity in glomerular mesangial cells	[166]
Myricetin	STZ-induced diabetic rats	antioxidant, antiinflammatory effect	(100 mg/ kg/day) for 6 months	- kidney tissues: alleviated DM-induced renal dysfunction, fibrosis, and oxidative damage and enhanced the expression of Nrf2 and its downstream genes, inhibition of the I-kappa-B (IκB)/nuclear factor-κB (NF-κB) (P65) is independent of the regulation of Nrf2.	[169]
Coenzyme Q10	Rats and/or mGECs	antioxidant effect	0.1% in the food for 7 weeks	- kidney tissue: restored the expression, activity and nuclear translocation of Nrf2 in high glucose-cultured mGECs, exerts beneficial effects in DN via mitophagy through restoring Nrf2/ARE signaling	[18]
AST	HF diet and STZ-induced diabetic rats	antioxidant effect, can delay DKD progression	25 mg/kg daily i.g. for 12 weeks	- kidney tissue: promoted the nuclear translocation of Nrf2 and increased its downstream protein HO-1 and SOD 1 expression, its reno-protective effect on DKD partly depends on Nrf2–ARE signaling.	[178]
Sodium butyrate	wild-type and Nrf2-knockout mice	Antioxidant effect	5 g/kg/day for 20 weeks	- kidney tissue: inhibited histone deacetylase (HDAC) activity and elevated Nrf2 and HO-1 and NAD(P)H dehydrogenase quinone 1, the expression of KEAP1, the negative regulator of NRF2, was not altered by its administration	[183]

Thio-2-propene-1-sulfinic acid S-allyl ester (Allicin); Systolic Blood Pressure Record (SBP); Kelch-like ECH associated-protein 1 (KEAP1); Nuclear factor erythroid 2-related factor 2 (Nrf2); Hypoxia inducible factor-1α (HIF-1α); Vascular endothelial growth factor (VEGF); Kidney injury molecule-1 (KIM-1); Glucose transporter 4 (GLUT4); insulin receptor substrates (IRSs); Grape Seed Proanthocyanidins Extract (GSPE); Streptozotocin (STZ); Superoxide dismutase (SOD); Mouse monoclonal antibody (HO-1); Glutathione S-transferase (GST); Eucommia ulmoides Oliv (EU); Advanced glycation end-product (AGE); Glyoxalase 1 (Glo1); Rho kinase (ROCK); Human renal glomerular endothelial cells (HRGECs); Cudrania tricuspidata Bureau (CTRE); The human renal proximal tubular cell line (HK-2); Murine glomerular endothelial cells (mGECs); Astaxanthin (AST); Human mesangial cells (HMC).

## Data Availability

Not applicable.

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
