# Peer review of "Oxidative Stress and NRF2/KEAP1/ARE Pathway in Diabetic Kidney Disease (DKD): New Perspectives"

_biomolecules, 2022, doi:10.3390/biom12091227_

Round 1
Reviewer 1 Report
This review discuss about the DKD mechanism, especially oxidative stress (OS) and NRF2/ARE/KEAP1 involvement, and highlight the potential of therapeutic approaches that target DKD via OS. I have minor concerns; thus, I suggest the authors address those concerns to make them more straightforward.
1. Line 162, “…… and glucose metabolism. [39,44–47]” should be replaced with “…… and glucose metabolism [39,44–47].”
2. Please check the references. The review must reflect the latest research on others, and if the quoted documents are obsolete references a few years ago, they can not reflect the latest research trends.
3. Please briefly explain the difference between DN and DKD.
4. Please choose the best NRF2 activators mentioned in the manuscript and talk about future application potential in terms of price and wide application.
Author Response
Dear Reviewer,
Firstly, we want to thank you in behalf of our team for your time and all your observations regarding our manuscript. We have taking in consideration your advices and made the following changes:
- We have checked and corrected as suggested.
- We have checked all the reference, and deleted as much as possible the older references. After revision we also, introduced the latest research in order to be up-to-date with our manuscript; that being said there are older references that remained as they were considered necessary and relevant to this field.
- The paragraph between line 67 and 70, explain the difference between DN and DKD as recommended.
- Line 675-688 we have included a paragraph in which we discuss the limitation of current OS strategies, characteristics, price and future potential wide application. To further improve this work, we have corrected any grammatical, and structural issues.

Reviewer 2 Report
Indeed, the topic addressed by a review manuscript by Daniela Maria Tanase et al., is of interest. However, the manuscript contains many flaws that make it hard to read or make sense of presented information. The authors should work hard in revising the manuscript to increase its quality to the level of publication. Some of the major flaws include, the lack of organization in presented information, there are many grammatical errors, while there is also no flow/cohesiveness of information. Many statements are without are without corresponding citation. lines 106-109, 155-157,..... etc.
Within the last paragraph of the introduction, it remains critical to indicate what information is covered by the review? So the reader can anticipate what is covered, and how....
Since the review is on interventions, what sources of information were targeted to retrieve information.
Briefly, mention why those bioactive compounds? The extent of information covered? Mention other relevant reviews on the topic, and mention how is the current review different, this will bring the significance of the current project.
Make your subheadings informative/objective and not abbreviated
The terms referring free radicals, like nitric oxide(NO•) and peroxynitrite (ONOO−), H2O2 or OH should be written appropriately…
In fact, the sections on the “DKD and hyperglycemia” or “DKD and inflammation” should be re-written, especially covering all the molecular mechanism discovered in the following sections.
Focus on molecular mechanisms and also mention biomarkers… those are different aspects.
In presenting results of previous studies its rather better to start with results from preclinical studies, before mentioning clinical data. The manuscript should supported by more clinical data.
If you are reporting on the therapeutic effects, mention the tissue or whether this is serum/blood… and what is the significance of that?
Although these bioactive compounds may present with potential therapeutic effects, but what considerations should be noted, or hindering development or approval for clinical use
Within table (2), mention the general effect of the bioactive compound before you mention makers, also stating where is this seen (tissue/blood)? Figure captions should also be revised, state where this seen, tissue or blood? And make captions complete, to stand alone by defining abbreviations.
Rewrite the abstract, stating the relevance of preclinical/clinical data in response to these bioactive compounds. What are the limitations in these antioxidants being effective?
Author Response
Dear Reviewer,
Firstly, thank you on behalf of our team for your time on peer-reviewing our manuscript. We have taken into consideration all your suggestion and comments, made structural changes, revised the whole manuscript for mistakes and readjusted the bibliography. Thus, we believe that with the help of your comments we improved the quality of this paper.
Within the last paragraph of the introduction, it remains critical to indicate what information is covered by the review? So the reader can anticipate what is covered, and how....
Since the review is on interventions, what sources of information were targeted to retrieve information.
Response: We followed your recommendation and introduced a new chapter “2. Materials and Methods” in which we described sources of information used to retrieve information, and selection criteria. Also, we considered appropriate to include here briefly the purpose of this review and the information covered.
In fact, the sections on the “DKD and hyperglycemia” or “DKD and inflammation” should be re-written, especially covering all the molecular mechanism discovered in the following sections.
Focus on molecular mechanisms and also mention biomarkers… those are different aspects.
Response: In order to complete these suggestions, we considered more appropriate to include data about hyperglycemia and inflammation as they are intricate mechanism, in the same section “3.1 The involvement of hyperglycemia and inflammation”. Thus, this section was rephrased and completed with information about other molecules involved, in correlation to the information described after, additional pathways and the potential biomarkers identified so far.
In presenting results of previous studies its rather better to start with results from preclinical studies, before mentioning clinical data. The manuscript should supported by more clinical data.
Response: Thank you for this observation, we have rearranged the information and sections such as start with preclinical studies, as recommended. We have introduced all the relevant human clinical studies identified. As see, therapeutic target such as bardoxolone methyl, is currently highly studied and therefore the higher number of clinical studies, compared to curcumin, sulforaphane or other natural compounds which hold results mainly from pre-clinical studies.
Briefly, mention why those bioactive compounds? The extent of information covered? Mention other relevant reviews on the topic, and mention how is the current review different, this will bring the significance of the current project.
Although these bioactive compounds may present with potential therapeutic effects, but what considerations should be noted, or hindering development or approval for clinical use
Response: To answer to these questions, at the end of therapy we have included two comprehensive paragraphs; First, line 675-688, in which we point out the limitation of the current known NRF2 activators and other OS-targets, and why their approval for clinical use is hampered, and other aspects regarding their wide-applications;
The second paragraph 689-709, includes the most relevant recent reviews of the last 2 years; we have researched again using electronic databases, with key words “NRF2, diabetic kidney disease, review”, 3 of them were not already mentioned in our bibliography (references listed below). We then identified the differences between them and our manuscript, and mentioned from our point of view how this paper can bring significance to the current project.
- Adelusi, T.I.; Du, L.; Hao, M.; Zhou, X.; Xuan, Q.; Apu, C.; Sun, Y.; Lu, Q.; Yin, X. Keap1/Nrf2/ARE Signaling Unfolds Therapeutic Targets for Redox Imbalanced-Mediated Diseases and Diabetic Nephropathy. Biomedi-cine & Pharmacotherapy 2020, 123, 109732, doi:10.1016/j.biopha.2019.109732.
- Gupta, A.; Behl, T.; Sehgal, A.; Bhatia, S.; Jaglan, D.; Bungau, S. Therapeutic Potential of Nrf-2 Pathway in the Treatment of Diabetic Neuropathy and Nephropathy. Mol Biol Rep 2021, 48, 2761–2774, doi:10.1007/s11033-021-06257-5.
- Tang, G.; Li, S.; Zhang, C.; Chen, H.; Wang, N.; Feng, Y. Clinical Efficacies, Underlying Mechanisms and Mo-lecular Targets of Chinese Medicines for Diabetic Nephropathy Treatment and Management. Acta Pharma-ceutica Sinica B 2021, 11, 2749–2767, doi:10.1016/j.apsb.2020.12.020.
If you are reporting on the therapeutic effects, mention the tissue or whether this is serum/blood… and what is the significance of that?
Within table (2), mention the general effect of the bioactive compound before you mention makers, also stating where is this seen (tissue/blood)? Figure captions should also be revised, state where this seen, tissue or blood? And make captions complete, to stand alone by defining abbreviations.
Response: We have introduced a new column and completed Table 2. with the general effects of the bioactive compounds, and in “Detection site and notable findings” column we have included where these results were seen, according to each study. Following your valuable comments, we have accordingly included a small paragraph in which we discuss de significance of therapeutic effects; Also, the figure captions were revised and completed as suggested.
Rewrite the abstract, stating the relevance of preclinical/clinical data in response to these bioactive compounds. What are the limitations in these antioxidants being effective?
Make your subheadings informative/objective and not abbreviated
Response: The abstract has been revised and completed with information, for a more robust overview of this review content. All the subheadings were rewritten and completed.
The terms referring free radicals, like nitric oxide(NO•) and peroxynitrite (ONOO−), H2O2 or OH should be written appropriately…
Response: Thank you very much for careful reading and thoughtful comments. We have checked, and corrected all the chemical form of the molecules mentioned. Also, we identified as much as possible all the grammatical issues, missing abbreviations, pointed and corrected them.

Reviewer 3 Report
The review discusses the role of oxidative stress and NRF2/KEAP1/ARE2 pathway in diabetic kidney diseases and discusses extensively several natural and synthetic therapeutic approaches. The review is very interesting and well-written. I would only recommend to rewrite the abstract with further details to be more representative of the review contents.
Another comment is that the table titles should be above them, while the full terms of the used abbreviations are placed in the table footnote.
Author Response
Dear Reviewer,
Firstly, thank you on behalf of our team for your time and recommendations.
As recommended, we have revised the abstract, rephrased and completed for a better overview of its content.
Thank you for this observation, the titles have been moved above, the abbreviations were completed, also, we corrected any other mistakes and missing information.

Round 2
Reviewer 2 Report
The revised paper in this present form has the potential to be published. Thanks for improving - manuscript gained quality and information.